# Does Cognitive Load Affect Measures of Consciousness?

**DOI:** 10.3390/brainsci14090919

**Published:** 2024-09-13

**Authors:** André Sevenius Nilsen, Johan Frederik Storm, Bjørn Erik Juel

**Affiliations:** 1Brain Signaling Group, Department of Physiology, Institute of Basic Medical Sciences, University of Oslo, 0372 Oslo, Norway; j.f.storm@medisin.uio.no; 2Vestre Viken Klinisk Nevrofysiologi, Kongsberg Hospital, Vestre Viken Health Trust, 3004 Drammen, Norway; b.e.juel@medisin.uio.no

**Keywords:** electroencephalography, consciousness, attention, signal diversity, perturbational complexity index, event-related potential

## Abstract

Background: Developing and testing methods for reliably measuring the state of consciousness of individuals is important for both basic research and clinical purposes. In recent years, several promising measures of consciousness, grounded in theoretical developments, have been proposed. However, the degrees to which these measures are affected by changes in brain activity that are not related to changes in the degree of consciousness has not been well tested. In this study, we examined whether several of these measures are modulated by the loading of cognitive resources. Methods: We recorded electroencephalography (EEG) from 12 participants in two conditions: (1) while passively attending to sensory stimuli related to the measures and (2) during increased cognitive load consisting of a demanding working memory task. We investigated whether a set of proposed objective EEG-based measures of consciousness differed between the passive and the cognitively demanding conditions. Results: The P300b event-related potential (sensitive to conscious awareness of deviance from an expected pattern in auditory stimuli) was significantly affected by concurrent performance on a working memory task, whereas various measures based on signal diversity of spontaneous and perturbed EEG were not. Conclusion: Because signal diversity-based measures of spontaneous or perturbed EEG are not sensitive to the degree of cognitive load, we suggest that these measures may be used in clinical situations where attention, sensory processing, or command following might be impaired.

## 1. Introduction

The ability to objectively detect the presence or apparent absence of consciousness in individuals is of clinical and theoretical relevance. Several methods aiming to objectively quantify the presence or degree of consciousness have been developed, such as the Glasgow Coma Scale [1], brain glucose consumption [2], and the electroencephalogram (EEG)-based BIS [3]. While many measures have shown promising results in distinguishing awake states from anesthetized to sleep states [4], it is important to also investigate the robustness of such measures within each state. For example, a robust clinical marker of consciousness should not be significantly affected by attention and other cognitive functions, which might be impaired or otherwise altered in patients. 

Some EEG-based measures are of particular interest because they can be performed at the bedside and are, at least conceptually, related to prominent theories of consciousness, which provides a degree of theoretical grounding. Such measures include the perturbational complexity index (PCI; [5]), measures of signal diversity of resting state EEG [6], and the P3b component of the sensory event-related potentials (ERP) elicited by the so-called auditory “global-local” oddball paradigm [7,8,9]. PCI is a measure based on quantifying the spatiotemporal complexity of the global EEG responses to local stimulation of the cortex (often by transcranial magnetic stimulation; TMS) and is inspired by the integrated information theory of consciousness [10]. The P3b ERP is a late EEG response elicited by the detection of irregularities in global temporal patterns and is inspired by the global neuronal workspace theory of consciousness [11,12,13]. While these two measures, in particular, have shown promise in distinguishing between coarse changes in the overall state of consciousness and have been used clinically [14,15], it is also important to ensure that they are not influenced by a variety of other changes within each state, provided that the overall state of consciousness remains unchanged.

Both attentional and cognitive load (i.e., task engagement) alter the underlying functional characteristics of the brain activity measured by EEG. For example, when attention is focused on certain sensory modalities, EEG shows increased activation in the related primary and secondary sensory areas [16], as well as in different sensory-related functional networks [17]. Similarly, task engagement is linked to changes in power spectral density distributions in EEG [18] and to changes in general task-related networks [19]. Such changes may affect measures of consciousness if the measures are sensitive to features of the EEG modulated by processes not related to consciousness. Therefore, as patients in clinical care cannot always be assumed to follow task instructions for a variety of reasons beyond being unconscious, it is important to verify that any measure of consciousness that is to be broadly clinically applicable is largely unaffected by changes in attentional focus and cognitive load. 

This study aimed to investigate whether changes in attentional and cognitive load cause modulation of PCI, the P300b event-related potential (P3b), and some EEG-based measures of signal diversity (conceptually similar to PCI) that have shown promise for classification of states of consciousness; Lempel Ziv complexity (LZc), synchrony coalition entropy (SCE), and amplitude coalition entropy (ACE) [4,6,20]. To examine whether changes in attentional and cognitive load modulation affect these measures, we implemented a within-subject design with a low-demand cognitive task related to the methodology behind the measures and a high-demand adaptive working memory. Based on the difference in nature of the various measures and prior findings, we hypothesized that only the P3b component of the evoked EEG should be affected by attentional and cognitive modulation.

## 2. Materials and Methods

### 2.1. Participants

12 healthy participants were recruited through written adverts and posters at the University of Oslo and Oslo University Hospital (*n_female_* = 7, age range 21–55 years, median 27.5 years). All participants received oral and written information and signed informed consent prior to this study. All participants could withdraw their consent at any point during this study and receive monetary compensation regardless of whether they completed this study or not. Inclusion criteria were age between 18 and 60 years, no magnetic resonance imaging (MRI) contraindications, no TMS-EEG risk factors [21], easily observable twitch responses to TMS stimulation over the thumb area in the motor cortex, and low level of visible activation of muscles on and around the head in response to TMS stimulation around target areas. This study was approved by the Regional Committee for Medical and Health Research Ethics (the Regional Committees for Medical and Health Research Ethics: 2014/1952-1).

### 2.2. Protocol and Tasks

For every participant, we performed a standardized MRI to allow for navigated TMS-EEG. Then we tested whether participants were likely to yield a strong TMS-evoked EEG response without muscle artifacts in the stimulation intensity range of our stimulator. This testing included a simple test with the TMS equipment where we aimed to find an approximate resting motor threshold (RMT, see below) and to investigate whether there was significant facial muscle activation when applying TMS pulses in the approximate target areas of interest. Participants visibly showed hand or finger twitches in response to TMS stimulation to the motor cortex (below 70% of the maximum TMS stimulator output), and little to no muscular artifacts from TMS pulses to target areas with an intensity set ~20% above the RMT were included in the main study. 

Once initial tests were completed, the EEG cap was mounted and electrodes were adjusted with ideal target impedances of <5 kOhm. Following setup, participants were seated in a reclining chair and completed 3 training trials of a working memory paradigm to minimize initial training effects and establish starting difficulty of the main experiment. We then performed the auditory experiments (for P3b), resting state EEG experiments (for LZc, ACE, and SCE), and the TMS-EEG experiments (for PCI) (order counterbalanced between participants). See Figure 1.

For the resting state experiments, we recorded EEG while participants completed 10 trials of the working memory task for the high-load distracted condition, and then for the low-load attentive condition, EEG was recorded for a similar duration while participants counted seconds passing based on the audible clicks from a nearby clock. In both conditions, the participants finally estimated how many seconds they believed had passed during the recording. 

For the TMS-EEG experiments, we first performed an automated RMT estimation to guide initial stimulator intensity. After finding a suitable location for stimulation, calibration of coil rotation, and stimulation intensity, the participant performed the distracted high-load condition while we were stimulated in the active phases of the task (5 pulses in the memorizing stage and 10 pulses in the maintenance phase). After 20 trials (300 pulses), we asked the participants to estimate how many pulses they had received. In the passive condition, we delivered a continuous train of 300 pulses (2 ± 0.3 s IPI) lasting for a total of 10 min.

For the auditory ERP experiments, 8 blocks of 105–135 trials were presented continuously. First, the auditory stimuli were presented while the participants solved the working memory task (an optional short break was given after 4 blocks), then later in the attentive low-load condition when participants were instructed to count the number of global deviants. To control those participants were in fact paying attention to auditory stimuli in the low-load condition, we asked participants after both conditions to indicate which of 10 specific auditory sequences they had heard and how certain they were (see Appendix A).

#### 2.2.1. Working Memory Task

In order to induce a high cognitive load, we designed a working memory task (Figure 2) where participants were instructed to memorize a string of letters (consonants) and then maintain the letters in working memory for 20 s before indicating which of three strings of three consonants were present in the original string. The two wrong alternatives included one consonant not present in the original string. The sequence of letters in the answers was randomized to minimize sequencing strategies. A trial started with a pause screen, which ended at a button press that initiated a 3 s countdown, followed by presentation of the text string for 10 s (update phase), fixation cross for 20 s (maintenance phase), and finally the three answer alternatives. The answer screen was present until a response was given. Consonants were chosen to increase difficulty (harder to “chunk”). In addition, to ensure high subjective cognitive load, the paradigm employed a staircase procedure so that for each correct answer, one additional letter (up to a total of 20) was added to the query string, while a wrong answer decreased the total number of letters by one (down to a minimum of 3). The text was white on a uniform gray background. The paradigm lasted for as long as necessitated by the measures.

#### 2.2.2. Low-Load Attentive Task

For each of the three methods (TMS-evoked, auditory-evoked, and spontaneous EEG) we employed a low-load paradigm consisting of merely paying attention to the stimuli. In the global-local auditory paradigm (for evoking the P3b), participants were asked to count the number of global deviants they heard and report the number after each block of stimuli. To increase generalizability across tasks, we employed similar aspects for the other methods (low-load attentive condition). For the TMS, we asked participants to count the total number of TMS pulses, and for the spontaneous EEG, we asked participants to count the number of seconds of audible clicks from an analog wall-mounted clock.

### 2.3. Equipment/Setup

Anatomical MRI recorded for TMS-EEG navigation purposes was scanned at Philips Ingenia 3T scanner (Intervention Center, University Hospital Oslo). The sequence employed was a T1 weighted sequence with 1.0 mm × 1.1 mm voxels with 184 slices and 2.0 mm thickness, reconstructed to 256 × 256 × 184 matrix, SENSE = 1.5 (RL), TR = 4.6 ms, TE = 2.3 ms, for a total sequence length of ~4 min. The sequence was preceded by a B1 calibration scan. 

For TMS stimulation we used a 70 mm figure-eight cooled coil (PMD70-pCool, MAG & More GmbH, Munich, Germany) together with a PowerMAG Research 100 stimulator (Mag&More, 81379 Munich, Germany). For 3D navigation, we used the NDI Polaris Spectra spatial navigation system employing two infrared cameras, motion trackers, and a 3D reconstruction of the participants’ head and brain (PowerMag view v1.7.4.401).

In order to calibrate the TMS stimulation intensity for each participant, the resting motor threshold (RMT) was estimated using a Mobi MINI TMSi Bluetooth-connected amplifier (MAG & More GmbH, Munich, Germany) in conjunction with PowerMag control software. Electrodes for the RMT device were fixed to the thumb of the dominant hand in order to record muscle activity from the abductor pollicis brevis muscle in response to single TMS pulses delivered to the thumb area of the contralateral motor cortex. RMT was estimated using an automatic algorithm that makes a maximum likelihood estimate of the stimulation intensity required to induce >50 mv peak-to-peak electromyographic (EMG) response to 50% of single TMS pulses [22].

EEG was acquired was acquired using equipment from Brainproducts GmbH (Gilching, Germany), with two BrainAmp DC 32 channel amplifiers and a 64 channel TMS compatible passive electrode cap (64Ch-EasyCap for BrainAmp using the 10–20 system of electrode placement, with ECI electrode gel and NuPrep abrasive skin-prep gel). The acquisition was performed using a 5 kHz sampling rate, DC-1000 Hz hardware filter, 16 bit rate, and ±16.384 mV measurement range., 

The working memory and global-local paradigms were programmed in Python 2.7.12 using PsychoPy2 (v1.82.02). The paradigms were run on a Dell Latitude 3550 laptop running Xubuntu 16.04. The working memory paradigm was presented on a screen approximately 140 cm in front of the participants’ head (FlexSan 2768, 19”, 60 hz, 1024 × 768, Eizo Nanao Corp., Ishikawa, Japan), with participants using a keyboard in their lap with keypad buttons “1”,“2”,“3” for each response alternative.

The stimulation protocol used for TMS-EEG for the PCI was similar to that of [5], with a series of 300 pulses (mean inter-pulse interval 2 [±0.3 s random jitter], full waveform), targeted at either the prefrontal cortex (BA6) or the parietal cortex (BA7) of the left hemisphere, with a stimulator intensity of between 120 to 160% of the RMT. Targeting was aided with 3D spatial navigation employing motion trackers, and 3D reconstruction of the participants’ skull and brain based on the participants’ own MR image. We adjusted intensity, location, and orientation of the induced magnetic field, to maximize the peak-to-peak amplitude of the initial deflection of the TEP (20–40 ms) and to minimize the level of artifacts present in the TEP. The goal was a minimum of 10 mV peak-to-peak amplitude in the electrodes close to the stimulation area, and minimal artifacts in all electrodes.

To mask TMS noise during TMS-EEG recordings, we used masking that has been generated from audio recordings of TMS coil clicks. A waveform containing multiple coil clicks of different intensity TMS pulses was chopped into ~1 ms subsections and then randomly shuffled in time (produced using custom scripts in Matlab M2017b). This generated a noise-like sound capable of masking the audible click from the TMS pulse at quite low volume. During the TMS experiments, the volume of the masking sound was set high enough (within comfort range) to completely mask the air-conducted sound generated by TMS pulses (the coil was held perpendicular to cortical stimulation). 

The auditory stimuli for the global-local paradigm were also generated in-house and consisted of 50 ms duration (including with 7 ms rise and fall time) chords of 3 sinusoids (either 350, 700, and 1400 Hz; or 500, 1000, and 2000 Hz—second and third partials were of 1/2 and 1/4 intensity, respectively). During the global-local experiments, the volume of the sounds was adjusted to what participants considered clearly audible speech level. Auditory stimulus was run on a Dell Latitude 3550 (Windows 8) using Sennheiser CX 2.00i earplugs (Sennheiser electronic SE & KG, Wedemark, Germany). 

All preprocessing and analysis were performed using Matlab (M2017b), employing in-house scripts and EEGLAB (v14.1.1).

### 2.4. Measures

#### 2.4.1. Evoked Measures

For measures based on ERPs and TMS evoked potentials (TEP), we computed P3b and PCI, respectively. PCI [5] requires repeated single pulses of TMS to locally perturb the cortex and a concurrent measurement of the following response with high-density EEG (producing the TEP). Then, a source reconstruction of the TEP is calculated, the inferred activations are binarized, and its complexity is estimated using LZc. In this way, PCI is intended to measure the degree of integration and differentiation in the brain—complex TEPs (high PCI) indicate integration and differentiation, while simple TEPs (low PCI) indicate that one or both aspects are diminished. 

The P3b marker, elicited by the global-local paradigm, measures the ERP of the late (>300 ms) EEG response to a break in global auditory patterns relative to habituated patterns (see [8]). To elicit the P3b, we employed an auditory paradigm (Figure 3) similar to that of [7]. Specifically, we presented 8 blocks of 105–135 trials. One trial consisted of five 50 ms tones with 150 ms inter-tone interval and an 1150 ms inter-trial interval. The first four tones were identical, with the fifth deviating or not, depending on trial conditions. A trial could be local deviant, which was defined as the fifth tone being higher or lower in frequency than the first four. The first 15 trials within a block were identical and either local deviant or not. This habituated a global pattern of local deviance. Global deviance was defined as a change in frequency of the fifth tone compared to the fifth tone in the majority (80%) of trials within a block (also the habituated trials). This resulted in four trial conditions with regards to how the local and global patterns could be disrupted: (1) local and global non-deviant, (2) local non-deviant and global deviant, (3) local deviant and global non-deviant, or (4) local and global deviant. For each participant, we presented 8 blocks (2 of each condition), lasting 30–40 min in total. Each block consisted of 15 habituation trials and 90–120 experimental trials, of which 20–30 were global deviant trials. The trials were semi-randomized to ensure a minimum of two 80% probability trials in between two 20% probability trials.

#### 2.4.2. Spontaneous Measures

To estimate signal diversity in the spontaneous EEG, we used measures of complexity (LZc) and coalition entropy (ACE, SCE). To estimate the complexity of the signal, we measured the compressibility of the discretized EEG signal through LZc using the 1976 variant [23] with algorithmic implementation by Kaspar and Schuster [24], and normalized by the compressibility of the shuffled (i.e., unstructured) data. To estimate the coalition entropy, we used ACE [6], a variant of the measure implemented by Shanahan [25] applied to discrete spontaneous EEG data (such as LZc) by calculating the entropy of the distribution of channel coalitions over time and normalizing with respect to the analytical maximum entropy. SCE [20] is similar to ACE, except that coalitions over channels are determined by whether pairs of channels are phase synchronized or not (difference between instantaneous phases of channel pairs smaller than 0.8 radians). 

LZc, ACE, and SCE have previously been used to separate conscious and apparently unconscious states, as well as distinguish between the normal and psychedelic conscious states [6,20,26,27,28,29]. 

### 2.5. Preprocessing

Generally, all data processing included standard steps—channel rejection based on visual inspection, downsampling, band pass filtering, re-referencing, independent component analysis (ICA) for ocular artifact correction (horizontal movement and blink artifacts), epoching with baseline correction, and epoch rejection—that were specifically adapted to the measures to be computed. See Table 1 for details.

While the preprocessing followed similar steps as in previous studies (PCI; [5], ACE, SCE, LZc; [6], P3b; [8]), some deviations should be mentioned. For the PCI measure, we first removed the pulse artifact (−1 to 10 ms around each TMS trigger) and replaced it with normally distributed white noise with mean and standard deviation of the baseline (−300 to −50 ms). For the spontaneous measures, we created non-overlapping windows of 5000 ms length and limited the number of electrode channels used in the analysis (after preprocessing) to 9 (F5, Fz, F6, C5, Cz, C6, P5, Pz, P6) to ensure that each 5 s window (3000 samples after downsampling) would have enough samples to estimate a probability distribution of 2^9^ = 512 possible states (required for SCE and ACE). The channels were selected to give broad topographical coverage. 

For all measures, we also implemented an automatic trial cleaning method where epochs were individually rejected using an automatic classifier based on a “noise” scoring algorithm:*S* = *m**_s_*(*CV* + *CD* + 1)
where *S* = score, *CV* = number of channels with over 200 μV peak-to-peak amplitude within the epoch, *CD* = number of channels with instantaneous amplitude change above 100 uV at least once in the epoch, and *m_s_* = mean of channel variances. This equation scores high with high variance, modulated by the number of channels showing unrealistic amplitudes or sharp deflections. Following the artifact scanning, we then rejected the 20% highest scoring of a subjects’ epochs (within condition) or all epochs over a cut-off determined by 20% of the group average, whichever was higher. The results of the automatic scanning algorithm were independently validated by two researchers performing manual classification on two randomly chosen subjects.

### 2.6. Analysis

#### 2.6.1. P3b

The epoched data from the global-local paradigm were analyzed following [8]. First, we averaged across trials according to the four trial types, depending on how the last tone in a trial related to the other tones in the trial and the global pattern in the block: Local Standard (LS), Local Deviant (LD), Global Standard (GS), and Global Deviant (GD). Based on the four trial types, the local and global effect was calculated as a t-statistic at each timepoint in the ERP of LD vs. LS and GD vs. GS, respectively. As we were only interested in the global effect (GD vs. GS), the local effect was not analyzed further. This resulted in an ERP of t-values for each channel and associated *p*-values. See Figure 4. 

To estimate the effect of global deviance, we thresholded the *p*-values according to the following rule: time points with a *p*-value that was (1) equal to or lower than the minimal *p*-value in baseline (−1000 to −200 ms) and (2) lower than 0.01 for at least 10 consecutive timepoints (25 ms) were coded as ‘0’ (‘1’ otherwise). Then, we averaged the values for the Pz channel as it covers the area of dominant response to global deviance [7] in the time window 300–700 ms after the onset of the fifth sound. This resulted in a single value for each subject, where a value of “1” indicated that there was no significant global effect according to the above criteria, and a value of < 1 indicated some significant global effect, with larger and longer lasting effects yielding scores close to 0. See Figure 4 (Fz channel shown for consistency).

#### 2.6.2. PCI

For PCI we computed a source reconstruction of each individual’s TEP using minimum norm estimate based on the MNI ICBM152 standard atlas MRI. Then the source activity was binarized based on estimating the significantly active sources by comparing their activity level to a bootstrap resampling of the maximum activation in the baseline activity. Briefly, a source was considered to be significantly active (and given a value ‘1’) at some point in time if its activity at that time was larger than the 99th percentile of the maximal amplitudes in the bootstrap resampled baselines. Otherwise, the source was considered to be ‘off’ at that time and was given a value ‘0’. This yielded a 2D binary matrix representing the spatiotemporal activity patterns of the cortical sources in response to a TMS pulse, from which the LZc was calculated on the interval 15–300 ms, concatenated in the spatial dimension. The complexity was then normalized based on the analytic formula for asymptotic peak LZc for a signal of a given length and source entropy to yield the PCI (see supplementary of [5] for further details). Single-trial responses are shown in Figure 5.

#### 2.6.3. Spontaneous EEG Measures

For the ACE and SCE, we first performed a Hilbert transformation and then binarized the data. For ACE, we used a median split of the amplitudes, setting any time point with amplitude above the median of that channel in the epoch to ‘1’ (‘0’ otherwise). For SCE, the thresholding was based on the synchrony between each pair of channels, so that a channel pair was considered to be in synchrony and given a value ‘1’ if the difference between their instantaneous phases was below 0.8 radians (‘0’ otherwise). The resulting binary matrices represented amplitude of activity for the set of channels over time (ACE), or the states of phase synchrony between channel pairs over time (SCE). Based on these matrices, we found the distribution of states over time, where a state (or ‘coalition’) was defined as the pattern of 1’s and 0’s across channels (ACE) or channel pairs (SCE) at one-time point. From the state distribution, we then calculated the sample entropy, normalized with respect to the entropy of shuffled sequences (maximum of 50 shuffles) of the same size and proportion of 1’s and 0’s as the original data [6]. Our final measure for the ACE was the mean ACE values over epochs (for each participant in each condition), while the final SCE value was the mean SCE over channels and epochs.

The LZc followed the same steps for binarization as for the ACE. However, here, the complexity was quantified by calculating the LZc (using the algorithm presented in Kaspar & Schuster, 1986). Each LZc value was normalized with respect to the complexity of the shuffled original data (maximum of 50 shuffles). The final value for the LZc for a participant in a condition was taken to be the mean LZc values across the epochs.

Examples of spontaneous resting-state data and their power frequency distribution are shown in Figure 6.

### 2.7. Statistics

We performed the same statistical analysis for all measures. First, we calculated the rate of true positives and false negatives according to thresholds for each measure (thresholds were estimated based on results reported in [5,6,8]). Since participants were considered to be conscious in both of our conditions, the ground truth for all the participants’ states was always positive (conscious). Thus, all classifications indicating unconsciousness (negative) were considered false negatives. Secondly, we performed pairwise *t*-tests to test whether there was any significant difference between the attentive and distracted conditions on the values of the different measures. Third, we implemented a simple data-driven classification algorithm to test if we could correctly classify the participants as being attentive (low-load) or distracted (high-load) based solely on the value of the measures recorded. Briefly, for each participant, we constructed distributions of each measure from all other participants in our sample. Then, we calculated the probability that the values observed for the given participant came from the attentive distribution: $$P=\frac{dA}{\left(dD+dA\right)}$$
where *P* is probability of datapoint coming from the attentive condition, *dA* is standardized distance from attentive condition distribution, and *dD* standardized distance from distracted condition distribution. This results in 0 < *P(A)* < 1, where *P(A)* > 0.5 indicates that the datapoint is more likely to come from the attentive condition, and *P(A)* < 0.5 indicates the opposite. Fourth, and finally, we calculated a ROC curve based on thresholding the probability distributions observed in point 4 to investigate whether a cut-off could be found that separated the two conditions. The last two analysis steps were made to test for the case that variance between the conditions was higher than the variance within conditions, which would lead to uncertainty in predicting the condition in which a single datpoint was recorded.

## 3. Results

Here we tested whether a high cognitive load (achieved by a demanding working memory task) affects a selection of proposed EEG-based measures of consciousness. The effects of performing a working memory task (high attentional load; distracted) versus not (low attentional load; attentive) on the various measures of consciousness can be seen in Figure 7 and Table 2. Specifically, we observed that for the ACE, PCI, SCE, and LZc, all participants scored higher than the pre-established thresholds of what is considered a positive indicator for the presence of consciousness (see [5,6,8] for the high-load distracted condition). In contrast, this was not the case for P3b. For the low-load attentive condition, all measures were above the threshold except for some participants who scored below the threshold for ACE (*n* = 3) and LZc (*n* = 1).

Additionally, we found no statistically significant difference between the high-load and low-load conditions for PCI, ACE, SCE, and LZc, whereas we found significantly higher P3b values for the high-load condition (*M* = 1.00 ± 0.00) than for low-load condition (*M* = 0.45 ± 0.34), *t* (11) = 5.62, *p* < 0.01 (Note that the test for P3b reduces to a one-sample *t*-test against the H0: mean = 1). Finally, P3b was the only measure that could be used to reliably separate the conditions on an individual basis using a threshold (Figure 7a) or a classifier (Figure 7c).

## 4. Discussion

In the present study, we tested whether attentional and cognitive load affected some recently proposed EEG-based measures of state of consciousness in humans. In summary, our results showed that the ACE, SCE, LZc, and PCI measures were not measurably affected by attentional or cognitive loading, suggesting that previous results are not substantially confounded by such effects [5,6]. In contrast, the P3b measure was strongly affected by attentional or cognitive load and showed a 100% ability to discriminate between the two conditions, an effect which is even more pronounced in our data compared to previous findings modulating attention [7,8]. This may suggest that the working memory task used here represents a stronger cognitive load than those previously employed. Supplementary analysis indicated that participants indeed focused their attention on the working memory paradigm and did not pay much attention to stimuli or duration (see Appendix A and Appendix B).

The present study was primarily designed as a quality control of the proposed measures, as any significant difference between the effects of cognitive load and passive attention to stimuli (or duration) would render the proposed measures less reliable tools for classifying states of consciousness (as opposed to measuring the content of conscious experience).

It should also be noted that, in the original studies employing the auditory “global-local” paradigm to elicit the P3b wave, the authors observed that participants doing a distracting task (or who were just allowed to mind-wander) during the paradigm did not show the same response as those that paid attention to the auditory stimuli [7,8]. The authors argued that the P3b can be more correctly described as an indicator of conscious access or attention to a stimulus, which is sufficient to infer the presence of consciousness. On the other hand, P3b might fail to be evoked due to lack of attention, faulty sensory processing, failure to follow commands, or disconnected awareness in the form of dreaming, as opposed to unconsciousness per se (see also [30]). Thus, in contrast to PCI and signal diversity measures, P3b should probably not be considered among candidate general markers of the state or level of consciousness, although the presence of a P3b response is a strong indicator of the presence of consciousness.

However, there are some limitations to our study. It could be argued that the low-load counting tasks did not only differ in the degree of cognitive load (counting vs. working memory) but also in the sensory domain (auditory vs. visual). It is, however, unlikely that activation of the different sensory domains would counteract the effect of the high-load working memory task. If anything, one would expect that a task vs. rest contrast would lead to decreased LZc during task performance [31]. We also re-analyzed data from a passive mind-wandering condition (*n* = 6) from a different experiment employing the same technical setup [32] and found similar values as presented in this paper, within ~1 SD of the distribution of the attentive, low-load condition (*M_ACE_* = 0.898, *M_SCE_* = 0.91, *M_LZ_* = 0.91). In addition, PSD analysis replicated the expected findings when comparing low and high-load conditions—increased theta band power frontal (Fz) during working memory modulation [18,33]—indicating that the high-load condition did indeed modulate working memory more strongly (see Appendix D). Note that earlier findings reported significant changes in frontal areas more broadly, whereas we only saw significant differences in the Fz channel. This might be because we included both encoding and retrieval parts of the high-load paradigm in the analysis. 

Secondly, for LZc and ACE in particular, we found some (*n* = 1/12 and 3/12, respectively) results below the conscious/non-conscious cut-off value employed previously [6]. However, the subjects were clearly conscious, and these values were relatively close to the thresholds compared to typical values observed during unconsciousness. These low values (relative to thresholds), may be related to three main factors: (i) differences in acquisition and preprocessing parameters [34], (ii) the selection of channels used for the final analysis, and (iii) the basis for threshold estimation. Regarding factors (i) and (ii), we deviate in our preprocessing steps from [6,20] when selecting the number of channels, the epoch length, and the sample rate. This is because one should not use a combination of channel number, epoch length, and sampling rate that would make it impossible to estimate the probability distribution of observed states in a maximum entropy state (the measures LZc, ACE, and SCE binarize the data in a channel by time matrix. As the number of unique states equals 2 channels one needs more samples than this in order to estimate the distribution of states within an epoch. Underestimating the distribution of states would lead to overestimating LZc, ACE, and SCE.) (see Appendix C). While [34] dismissed this issue, we could not find any strong theoretical or empirical reason to do so. Regarding factor (ii), a more rigorous investigation of the complexity as a function of the specific channel selection is required to understand the effect. However, it is conceivable that the subset of channels chosen affects the absolute values of the measures enough to push them under the threshold. Regarding factor (iii), it seems possible that differences in demographic variables, technical and practical implementation, or data cleaning and/or analysis could warrant that a new cut-off should be calculated, based on data gathered locally. Thus, the exact results from these three methods may depend on details of equipment, preprocessing, and various confounding variables. 

Third, a sample size of 12 is on the lower end. However, the replication of measures LZc, ACE, and SCE in an independent sample doing mind wandering (*n* = 6), as mentioned above, gave similar values as currently presented. In addition, the mean values observed were within 1 standard deviation of each other and well above previously recorded thresholds (see Table 2). This suggests that the means would not change with a higher sample and that even if the difference would be significant, it is questionable whether the difference would be of relevance. However, our sample is relatively young, and older adults may employ different cognitive strategies and functional networks for problem-solving [35], limiting generalizability.

Finally, while the present study did not employ a non-conscious control, there is a need for validating measures within conscious states to control for possible confounding factors such as muscle tension and memory. Muscle tension in particular has been shown to influence EEG recordings [36] and result in EEG-based metrics of anesthetic depth, such as the bispectral index, producing false negatives [37]. Moreover, attenuation of memory formation or recall caused by anesthesia may be confounded with loss of consciousness, as assessed by post-anesthesia reports (for discussions of recall and unconsciousness in anesthesia, see [30,38]. To control for such effects, proposed measures of conscious state can be investigated in, for example, paralysis caused by muscle relaxants (as in [36]), or in patients with abnormal muscle tension due to amyotrophic lateral sclerosis (ALS) [39], or in conscious patients with anterograde amnesia as in Korsakoff’s syndrome [40].

In summary, our study showed that four candidate measures of consciousness—PCI, ACE, SCE, and LZc—were not measurably affected by attentional and cognitive load and are thus not dependent on the subjects’ task compliance or direction of attention. While there are limitations to this study, no significant difference between conditions can be taken to mean (A) no difference of relevance between conditions for PCI, LZc, ACE, and SCE, or (B) an actual difference confounded by a counteracting effect such as loading of different sensory modality. Given the observed difference in frontal theta power (Appendix D) and the purported association between measures of complexity and consciousness, we are partial to hypothesis A.

We suggest that future studies of measures of consciousness should not only focus on the fully conscious vs. fully unconscious divide but also further test influences from possible confounding factors, including changes in muscle tone or memory, as well as altered states of consciousness. This may help identify candidate measures that are suitable for both clinical and basic research points of view.

## Figures and Tables

**Figure 1 brainsci-14-00919-f001:**
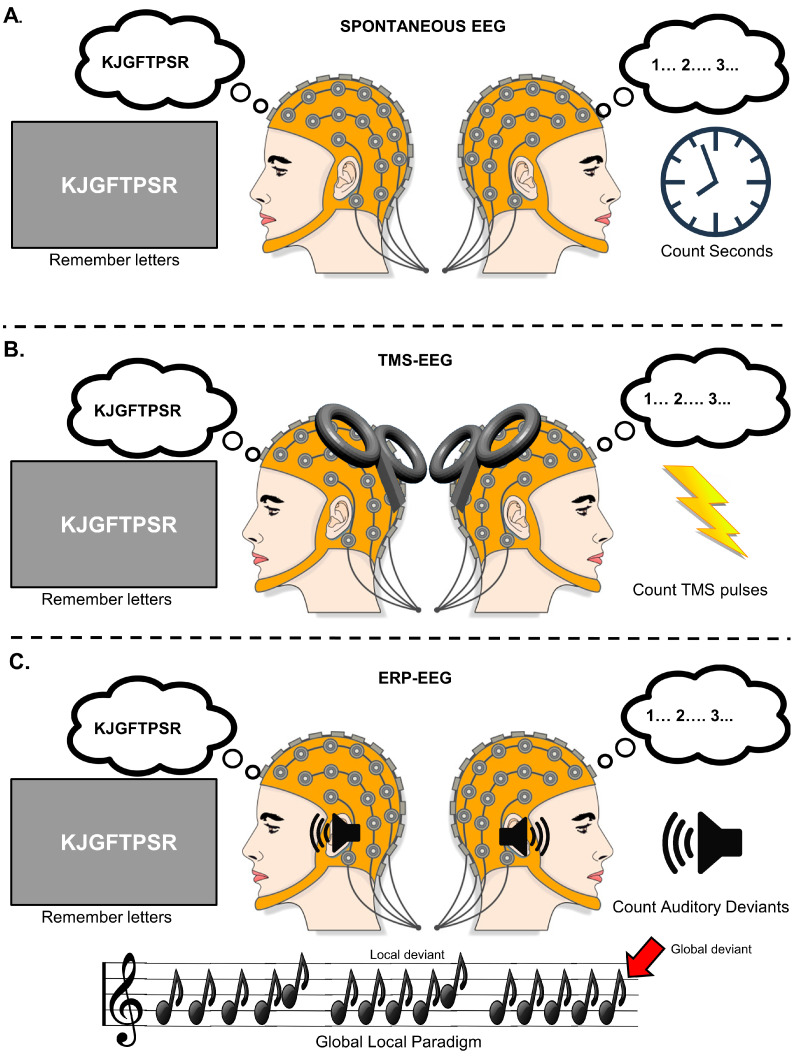
Overview of experimental protocol. Each experiment consisted of three pseudo-randomized blocks with two conditions (low-load stimuli attentive, high-load working memory). (**A**) Spontaneous EEG recording while receiving and counting audible seconds and while performing working memory task. (**B**) TMS-EEG recording while receiving and counting TMS pulses, and while performing working memory task. (**C**) ERP-EEG recording while receiving and counting number of global auditory stimuli pattern deviations in the global-local paradigm and while performing working memory task simultaneously.

**Figure 2 brainsci-14-00919-f002:**
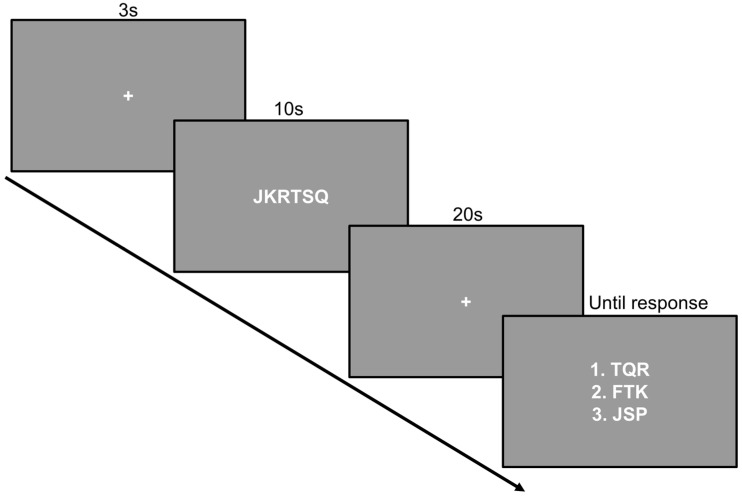
Working memory paradigm employed for attentional and cognitive load. One trial consisted of first a fixation cross, then a string of consonants (update phase), fixation cross (maintenance phase), and then three response alternatives (response phase). The answer options consisted of three consonants each, where the two wrong alternatives contained one incorrect letter (not present in the original string).

**Figure 3 brainsci-14-00919-f003:**
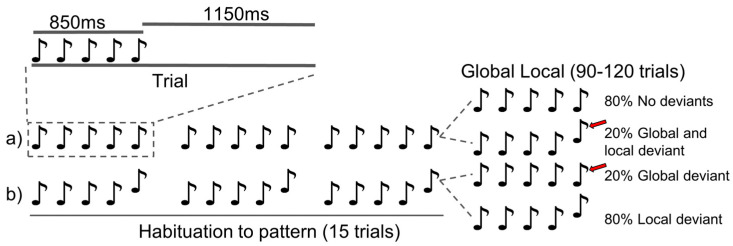
The “global-local” paradigm. The paradigm consisted of 8 blocks of 105–135 trials, where each trial had five tones, with the fifth tone deviating or not from the previous four. The first 15 trials of a block were habituation trials and followed the same pattern: (**a**) no deviant fifth tone, or (**b**) deviant fifth tone. The next 90–120 trials had a 20% chance of deviating from the pattern established in the habituation part of the block (red arrow).

**Figure 4 brainsci-14-00919-f004:**
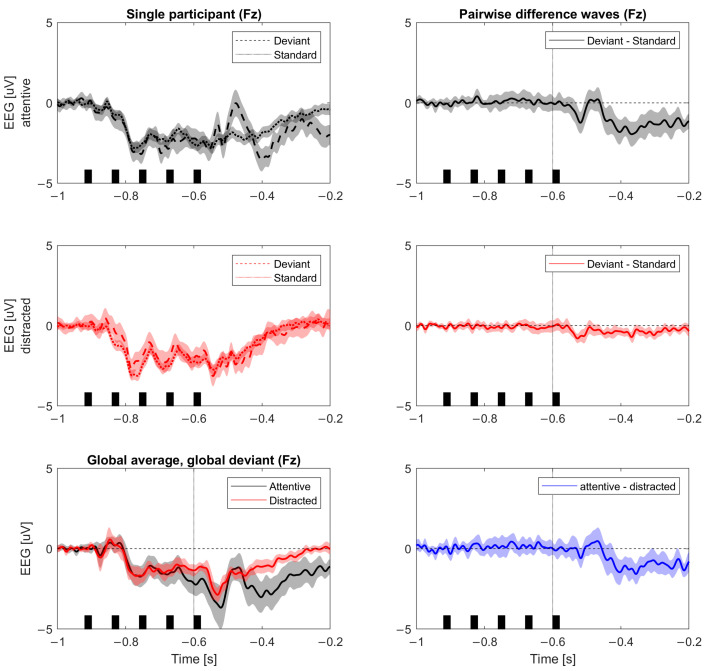
Auditory evoked data. Evoked potentials for channel Fz for one participant. Raw values for each condition on the left and their difference on the right. Panels show the evoked response to the 5th tone (black bars) either following the pattern of preceding trials (standard) or deviating from it (deviant). (**Top**) panel represents the attentive condition in which participants pay attention to the 5 tones; (**Middle**) row represents the distracted condition in which participants perform a working memory task; and the (**Bottom**) panel compares the top and middle panels. The bottom right panel represents the basis for computing P3b (the evoked response after 300 ms).

**Figure 5 brainsci-14-00919-f005:**
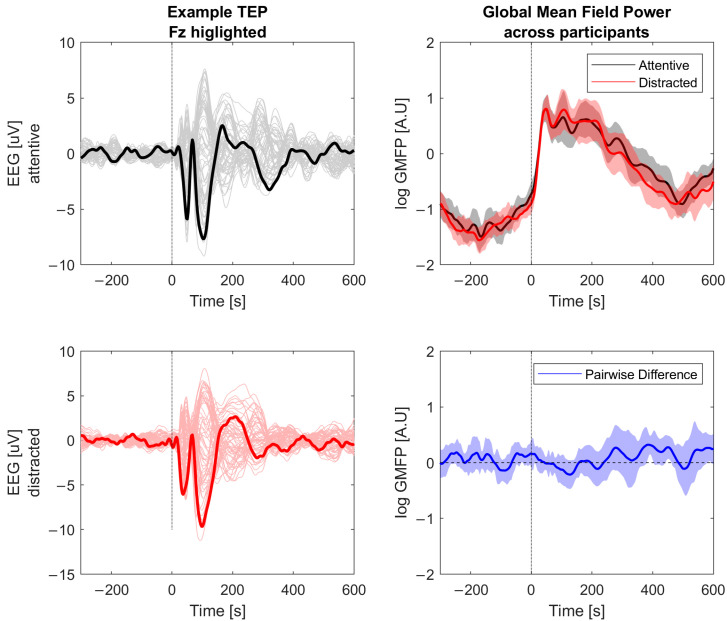
TMS-EEG data. Left panels show the TMS evoked potential (TEP) in the attentive (**top**) and distracted (**bottom**) conditions for all EEG channels with Fz highlighted. This is the basis for calculating PCI. Right panels show the global mean field power of the TEP (**top**) for both conditions and their difference (**bottom**).

**Figure 6 brainsci-14-00919-f006:**
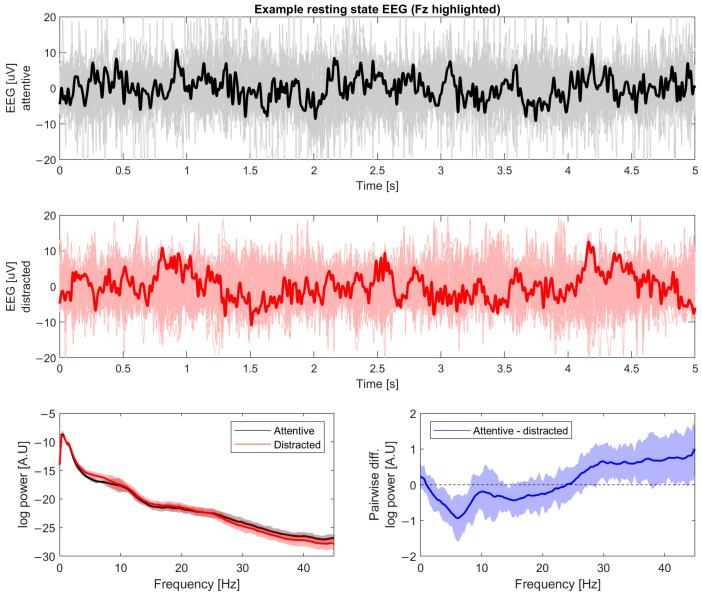
Resting state EEG data. (**Top**, **Middle**) panels show an example epoch of spontaneous resting-state data recorded in the attentive condition and distracted condition, respectively. This is the basis for calculating signal diversity measures. (**Bottom**) panel shows their power frequency distribution of both conditions (**Left**) and their difference (**Right**).

**Figure 7 brainsci-14-00919-f007:**
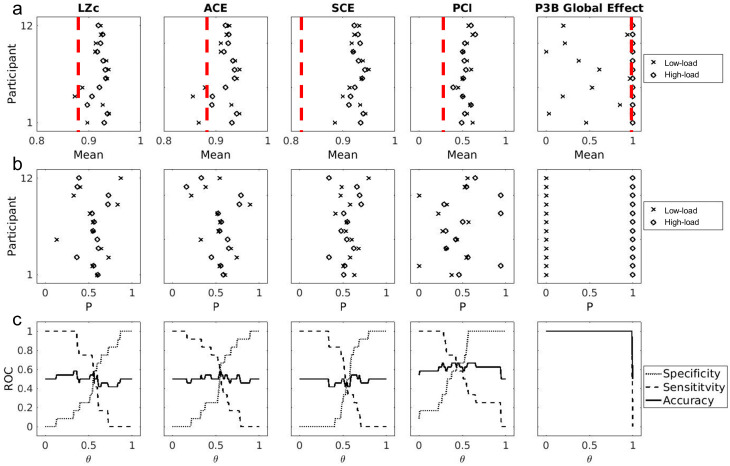
Effects of attentional load on various measures of conscious states: overview of results. Results for each measure for the distracted (high-load) and attentive (low-load) conditions. (**a**) Mean score for each measure under each condition for each participant. The dashed red lines indicate thresholds for positive indicators of presence of consciousness, extracted from previous publications [5,6,8]. (**b**) Probability of being classified as “attentive condition” given the overall distribution of the group for each participant in each condition. (**c**) Sensitivity, specificity, and accuracy of varying linear thresholds θ based on the probability distributions in panel (**b**).

**Table 1 brainsci-14-00919-t001:** Overview of preprocessing steps.

P3b Auditory Evoked Potential	PCI TMS-EEG	LZc, ACE, SCE Spontaneous EEG
Not conducted	Remove and replace pulse artifact [−1, 10] ms	Not conducted
Remove and interpolate bad channels		
Downsampling: 250 Hz	362.5 Hz	600 Hz
Bandpass filter: 0.5–100 Hz	0.1–45 Hz	0.5–100 Hz
Average reference		
ICA decomposition		
Ocular artifact ICs removed		
Lowpass filter: 20 Hz	Not done	45 Hz
Epoched: −1000–1000 ms around 5th sound	−300–600 ms around TMS pulse	0–5000 ms non-overlapping
Baseline subtraction: −1000–−50 ms	−300–−50 ms	Linear detrend
Epoch rejection		
Not conducted	ICA decomposition	Channel selection
Not conducted	Pulse-related ICs removed	Not conducted

Measures and their respective preprocessing steps. Removal of bad channels consisted of manual inspection, with a focus on dead channels or channels with consistent high-amplitude artifacts relative to the other channels. Removed channels were interpolated. Independent Component Analysis (ICA) was used to decompose the signal into independent components (ICs), which were manually inspected and removed if they appeared to be associated with ocular or pulse-related artifacts. Epoching was centered on the TMS pulse, the fifth auditory stimuli within a series (see analysis), or continuous non-overlapping windows, depending on method. Baseline subtraction consisted of removing the average amplitude of the specified baseline relative to the epoch center/zero point.

**Table 2 brainsci-14-00919-t002:** Descriptive statistics and results for each measure.

	Mean ± SD					Above Threshold		Classifier
	Low-Load	High-Load	*t*(11)		*θ*	Low-load	High-Load	Accuracy
LZc	0.92 ± 0.02	0.92 ± 0.01	−0.94		>0.88	~92%	100%	0.58
ACE	0.91 ± 0.03	0.92 ± 0.02	−1.03		>0.89	75%	100%	0.58
SCE	0.92 ± 0.02	0.93 ± 0.01	−0.74		>0.82	100%	100%	0.54
PCI	0.55 ± 0.05	0.53 ± 0.07	1.59		>0.31	100%	100%	0.67
P3B	0.45 ± 0.34	1.00 ± 0.00	−5.62	**	=1	100%	0%	1.00

The table shows mean ± standard deviation (SD) for each measure (Ms) in terms of attentive (low-load) and distracted (high-load) conditions, with t score for pairwise Student’s *t*-test, previously reported thresholds θ, observations above threshold θ (low-load | high-load), and peak accuracy using a means distribution classifier. **; *p* < 0.01, LZc: Lempel Ziv complexity, ACE; amplitude coalition entropy, SCE; source coalition entropy, PCI; perturbational complexity index, P3B; the late (>300 ms) evoked positivity of deviance from established pattern induced by the ‘global-local paradigm’.

## Data Availability

The raw data supporting the conclusions of this article will be made available by the authors upon request.

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
