# Peer review of "Does Cognitive Load Affect Measures of Consciousness?"

_brainsci, 2024, doi:10.3390/brainsci14090919_

Round 1

Reviewer 1 Report

Comments and Suggestions for Authors

The researchers enlisted a cohort of 12 healthy individuals who were subjected to electroencephalographic recordings under distinct experimental conditions, with the objective of ascertaining potential variations in EEG-derived metrics between the said conditions. Howerver, there are certain issues that need to be addressed in this article.

1.A sample size of 12 participants is quite small, and hence may not provide sufficient statistical power necessary to detect significant differences between the conditions. The authors should consider the implications of this on their results.

2. The low-load condition involved a simple counting task, which differs from the high-load condition not only in cognitive demand but also in the nature of the task (auditory vs. visual). This difference probably could have cuckolded the findings.

3.Several research have shown that older adults tend to utilize different functional brain networks and processing strategies than younger adults when required to engage in demanding cognitive activities because of low cognitive efficiency and the availability of compensatory mechanisms. The generalizability of the study's findings to the broader middle-aged and elderly population may be constrained.

Ref:

Brain reserve affects the expression of cognitive reserve networks. Hum Brain Mapp. 2024;45(5):e26658.

Moderating effects of cognitive reserve on the relationship between brain structure and cognitive abilities in middle-aged and older adults. Neurobiol Aging. 2023;128:49-64.

4.The authors should provide a more balanced view by discussing the potential shortcomings and the need for further research to validate their conclusions.

Author Response

We thank the reviewer for their comments and insights. We have answered each comment below, with references to the manuscript where changes have been made (and the relevant change in quotes).

Major comments:

  1. A sample size of 12 participants is quite small, and hence may not provide sufficient statistical power necessary to detect significant differences between the conditions. The authors should consider the implications of this on their results.

-        Indeed, a sample of 12 is on the lower end to detect small effect sizes. However, we argue that an increased sample size would at best result in a significant difference with an effect size too small to matter. This is due to three reasons: A) we replicated the findings for the low load condition in a mind-wandering condition (with measures LZc, ACE, SCE) in a different dataset (with same recording setup) with a sample size of 6. This suggests that, at least for the low load condition, the mean would stay close to the same even with increased sample size. B) the mean of the high load condition was within 1 standard deviation of the mean of the low load condition, for all measures except the P3b, which suggests that even if a significant difference can be detected with an increased sample size, the effect size is small enough that we wouldn’t be comfortable calling it relevant. C) All values were well above previously established thresholds for inferring consciousness (or absence thereof), which would mean that any significant finding (given that the mean remains within ~1 standard deviation of current results) would be of little clinical relevance. The only exception is the P3b measure which was significantly different for every single individual—in other words, increasing sample size is unlikely to make this finding non-significant.

-        We have added the following to the discussion (lines 513-519): “Third, a sample size of 12 is on the lower end. However, the replication of measures LZc, ACE, and SCE, in an independent sample doing mind wandering (n = 6), as mentioned above, gave similar values as currently presented. In addition, the mean values observed were within 1 standard deviation of each other and well above previously recorded thresholds (see Table 2). This suggests that the means would not change with a higher sample and that even if the difference would be significant it is questionable whether the difference would be of relevance.”

  1. The low-load condition involved a simple counting task, which differs from the high-load condition not only in cognitive demand but also in the nature of the task (auditory vs. visual). This difference probably could have cuckolded the findings.
    1. We addressed this difference in the discussion (lines 474-478), although it may not have been sufficiently clear or exhaustive. The replication of the analysis (LZc, ACE, SCE) in a different dataset (with same recording setup) employing a mind-wandering task (lines 486-488) suggests that differing processing modalities (here, presumably the default mode network is active) does not alter the results. If anything, we would expect that the high load condition might decrease LZc [1]. 
    2. We’ve highlighted this in the discussion section (lines 479-484): “If anything, one would expect that a task vs. rest contrast would lead to decreased LZc during task performance [42]. We also re-analyzed data from a passive mind wandering condition (n=6) from a different experiment employing the same technical setup [32], and found similar values as presented in this paper, within ~1 SD of the distribution of the attentive, low-load condition (MACE=0.898, MSCE=0.91, MLZ=0.91). 
      1. [1]/[42] Northoff, G. and Gomez-Pilar, J. (2021) ‘Overcoming rest-task divide-abnormal temporospatial dynamics and its cognition in schizophrenia’, Schizophrenia bulletin, 47(3), pp. 751–765.

  1. Several research have shown that older adults tend to utilize different functional brain networks and processing strategies than younger adults when required to engage in demanding cognitive activities because of low cognitive efficiency and the availability of compensatory mechanisms. The generalizability of the study's findings to the broader middle-aged and elderly population may be constrained.
    1. We have now added a line that mentions this limitation: “However, our sample is relatively young, and older adults may employ different cognitive strategies and functional networks to problem solving [43], limiting generalizability.
      1. [43] Brain reserve affects the expression of cognitive reserve networks. Hum Brain Mapp. 2024;45(5):e26658.
  2. The authors should provide a more balanced view by discussing the potential shortcomings and the need for further research to validate their conclusions.
    1. We have now increased the limitation section slightly to highlight these issues of generalizability (lines 541-546):While there are limitations to the study, no significant difference between conditions can be taken to mean A) no difference of relevance between conditions for PCI, LZc, ACE, and SCE, or, B) an actual difference confounded by a counteracting effect such as loading of different sensory modality, Given the observed difference in frontal theta power (Appendix D) and the purported association between measures of complexity and consciousness, we are partial to hypothesis A.

Reviewer 2 Report

Comments and Suggestions for Authors

This is a very interesting study in which the authors studied how measures of consciousness are affected by the loading of cognitive resources. They conducted a combined EEG-TMS study and found that an auditory-evoked event-related potential is significantly altered when simultaneously performing a working memory task. 

The paper is well written, and the rationale is clear. The results are correctly discussed, and they support the conclusion. I have a few methodological questions which hopefully help the authors to further strengthen their manuscript. 

The authors derive t- and p-values for the ERPs for every channel (lines 319-323). They threshold the p-values according to a few objective criteria but I wonder how this method deals with the multiple comparison problem and whether their method may result in false positives. One way to deal with the multiple comparison problem is using cluster-based approaches together with permutation testing (see for example Maris & Oostenveld [2007]). I was wondering whether the authors can confirm that they get similar results in qualitative sense when they rerun their statistical analysis using an approach such as cluster-based permutation testing.  

Finally, can the authors provide information on how many independent components were removed during the manual rejection (lines 276-280)?

References

Maris, E., & Oostenveld, R. (2007). Nonparametric statistical testing of EEG-and MEG-data. Journal of Neuroscience Methods164(1), 177-190.

Author Response

We thank the reviewer for their comments and insights. We have answered each comment below, with references to the manuscript where changes have been made (and the relevant change in quotes).

Major comments: 

  1. The authors derive t- and p-values for the ERPs for every channel (lines 319-323). They threshold the p-values according to a few objective criteria but I wonder how this method deals with the multiple comparison problem and whether their method may result in false positives. One way to deal with the multiple comparison problem is using cluster-based approaches together with permutation testing (see for example Maris & Oostenveld [2007]). I was wondering whether the authors can confirm that they get similar results in qualitative sense when they rerun their statistical analysis using an approach such as cluster-based permutation testing.  
    1. While the reviewer is correct that a lack of some sort of multiple comparison correction will result in inflated false positives, we will not implement a correction at this point for five reasons: A) we only use the Pz channel in the analysis (line 328), B) we already implement some degree of cluster based thresholding (line 326), C) our method follows closely that of previous studies, and a significant deviation from established methodology would weaken the generalizability of our results wrt. P3b to the clinic, D) our study replicates previous findings making stricter statistics unnecessary, and, E) given that we average the binarized p-values, false positives would give the P3b method the benefit of the doubt in terms of the high load condition, yet the fact the high load condition showed no P3b response (not even a false positive) suggests that if anything, we should be more lenient than we already are. That is, we should try to find a P3b response during cognitive load when there is none (although we didn’t find a response) rather than increasing the chance of finding no response when there is (a much worse alternative ethically speaking given that measures like the P3b is used clinically to infer the presence of consciousness).
  2. Finally, can the authors provide information on how many independent components were removed during the manual rejection (lines 276-280)?
    1. Two ICA components were removed, specifically those corresponding to horizontal eye movements and blinks. We have added this (line 278-279): “independent component analysis (ICA) for ocular artifact correction (horizontal movement and blink artifacts)

Reviewer 3 Report

Comments and Suggestions for Authors

I thought this paper provides a useful contribution to the literature in comparing multiple methods of measuring consciousness. The effect of cog load on the P300b event potential was an interesting finding compared to the other measures. 

n=12 limits the strength of the paper's conclusion -- a slightly large sample size could have been advantageous for better arguing for the generalization of the paper's results.

I am interested in the salience network which most likely related to the effects the authors were measuring. It would have been interesting to see a line or two related to this subject.

Author Response

We thank the reviewer for their comments and insights. We have answered each comment below, with references to the manuscript where changes have been made (and the relevant change in quotes).

Major comments:

  1. n=12 limits the strength of the paper's conclusion -- a slightly large sample size could have been advantageous for better arguing for the generalization of the paper's results.
    1. We agree that a higher sample size would have been beneficial, although we don’t see trends that would indicate that there would be a significant finding of relevance even with a larger sample. Especially given the replication of LZc, ACE, and SCE in a mind wandering condition (lines 486-488), the small difference in means (< 1 standard deviation), and that none of the measures (except P3b) scored below previously established thresholds. Thus, even if a significant effect could be observed with a larger sample, it would likely be very small and not clinically relevant in terms of inferring consciousness in patients. However, we have raised this issue more prominently in the limitations section (lines 517-523): “Third, a sample size of 12 is on the lower end. However, the replication of measures LZc, ACE, and SCE, in an independent sample doing mind wandering (n = 6), as mentioned above, gave similar values as currently presented. In addition, the mean values observed were within 1 standard deviation of each other and well above previously recorded thresholds (see Table 2). This suggests that the means would not change with a higher sample and that even if the difference would be significant it is questionable whether the difference would be of relevance.
  2. I am interested in the salience network which most likely related to the effects the authors were measuring. It would have been interesting to see a line or two related to this subject.
    1. While a thorough discussion of how activation of the various resting and task related functional networks contribute to e.g. neural complexity (as estimated by Lempel Ziv complexity) is outside the scope of the current article, there are a few findings that may suggest there is a connection – although would require a lengthy discussion and review. For example, there have been noted differences in rest vs. task in terms of Lempel Ziv complexity where LZc is observed to decrease during task performance [1]. In terms of our study it could be taken to mean that LZc is artificially low in both the low and high load conditions as the low load condition is not a pure resting state or mind wandering condition. Further, meditation is seen to increase LZc vs. rest [2] and some studies see that e.g. the salience network is upregulated during meditation  [3]. In terms of our study, given that we see no difference, then one can assume that to the extent the salience network is involved it is either not upregulated in one condition over the other, or that some other factor counteracts it. While a discussion of this would be interesting, the space of potential factors involved would be too large for the current findings. The same caveat applies to the hypothesis that various resting state and task networks may be associated with different neural complexity. Our study is not equipped to elucidate such a relationship without assumptions about which networks are specifically up/downregulated when. We have however added a note to the article regarding the rest vs task difference in LZc discussed in [1] as that is the most pertinent finding in relation to the current paper (lines 479-480):If anything, one would expect that a task vs. rest contrast would lead to decreased LZc during task performance [42].” 
      1. [1]/[42] Northoff, G. and Gomez-Pilar, J. (2021) ‘Overcoming rest-task divide-abnormal temporospatial dynamics and its cognition in schizophrenia’, Schizophrenia bulletin, 47(3), pp. 751–765.
      2. [2] Atad, D. A., Mediano, P. A. M., Rosas, F., & Berkovich-Ohana, A. (2023, June 28). Meditation and Complexity: a Systematic Review. https://doi.org/10.31234/osf.io/np97r

[3] Vishnubhotla, R. V., Radhakrishnan, R., Kveraga, K., Deardorff, R., Ram, C., Pawale, D., ... & Sadhasivam, S. (2021). Advanced meditation alters resting-state brain network connectivity correlating with improved mindfulness. Frontiers in Psychology, 12, 745344.

Reviewer 4 Report

Comments and Suggestions for Authors

In the paper titled "Does Cognitive Load Affect Measures of Consciousness?", I have some key suggestions for improvement, questions for clarification, and highlighted language corrections:

The paper describes the use of EEG and TMS but lacks details regarding the calibration of equipment and the selection of electrode placements. Further clarification on why specific EEG markers (P3b, PCI, etc.) were selected would provide a stronger rationale for the study.

You mention that participants performed a demanding working memory task. How was the cognitive load calibrated to ensure consistency across participants? Were there any differences in the subjective cognitive load experienced by different individuals?

Why were the specific EEG channels (Fz, Cz, etc.) chosen for this study? Were other channels considered, and if so, why were they excluded?

Given that cognitive load affects certain EEG markers, how do you suggest this study can be translated into clinical practice? Specifically, in patients who might not be able to perform cognitive tasks, how would this impact the use of EEG for measuring consciousness?

These revisions and additional clarifications will help improve the overall clarity, methodological rigor, and clinical relevance of the paper.

Comments on the Quality of English Language

In the sentence "Several methods aiming at objectively quantifying the presence or degree of consciousness have been developed", the phrase "aiming at" should be rephrased to "aiming to" for better clarity.

Author Response

We thank the reviewer for their comments and insights. We have answered each comment below, with references to the manuscript where changes have been made (and the relevant change in quotes).

Major comments:

  1. The paper describes the use of EEG and TMS but lacks details regarding the calibration of equipment and the selection of electrode placements.
    1.  We have now added the channel distribution (10-20 system) (lines 191-192): “using the 10-20 system of electrode placement.“ The equipment otherwise hasn’t been calibrated beyond ensuring <5kOhm impedance (line 101) and setting the TMS to 120-160% (individually adjusted to maximize evoked response and minimize artifacts) (line 204 and 207-209), audible tones in the auditory paradigm (line 223), visible screen for the working memory paradigm (lines 195-197), and so on. If there are calibration details missing, please let us know which you would like presented.
  2. Further clarification on why specific EEG markers (P3b, PCI, etc.) were selected would provide a stronger rationale for the study.
    1. The specific EEG markers were selected based on their use in the previous literature, as well as what they aim to capture. Specifically, P3b and PCI were selected as they have been widely studied, are both perturbational (evoked), yet seldom compared (lines 40-52,68,69). We knew from beforehand that P3b was likely to be affected by distraction (e.g. a working memory task) while no such study had been performed on PCI. These two measures are also associated with two dominant theories of consciousness (PCI; integrated information theory, P3b; global workspace theory). The spontaneous measures LZc, ACE, and SCE, were selected based on their prior use, and their conceptual similarity to PCI (i.e. measuring neural differentiation) although not perturbational in nature. This latter point has been highlighted on line 67: “(conceptually similar to PCI) Spontaneous measures are also more likely to be applicable in the clinic. There are a multitude of other measures that could have been chosen, but many measures correlate with each other (unsurprising if they measure the same underlying dynamics). In Appendix D, we also have a spectral power comparison. Together, these measures cover most of the categories of types of measures used in consciousness science.
  3. You mention that participants performed a demanding working memory task. How was the cognitive load calibrated to ensure consistency across participants? Were there any differences in the subjective cognitive load experienced by different individuals?
    1.  The working memory paradigm was calibrated with a staircase method (as described in lines: 149-152). This ensures that the difficulty of the working memory task was in principle constant over time (at the 50/50 correct/wrong threshold), although errors or lucky guesses may shift the difficulty beyond +- 1 step. The calibration was not only done throughout the study, but also prior to the experiment in order to set the difficulty close to the individual 50/50 threshold (line 104). While we did not investigate whether participants subjectively experienced the task to be more or less difficult, the staircase method is an established method for calibrating difficulty in such tasks.
  4. Why were the specific EEG channels (Fz, Cz, etc.) chosen for this study? Were other channels considered, and if so, why were they excluded?
    1. EEG channels selected for analysis were based on previous studies. P3b is most prominent over posterior-central channels corresponding to channel Pz (added “it covers the area of dominant response to global deviance [7]” to line 329), PCI employed all channels, and LZc, ACE, and SCE, used 9 channels which were selected based on their spatial coverage (i.e. maximizing spread while keeping distance between electrodes relatively uniform) (lines 297-298). However, the Fz channel is shown in the plots for consistency across measures. A note on this has been added to line 334: “See Figure 5 (Fz channel shown for consistency).
  5. Given that cognitive load affects certain EEG markers, how do you suggest this study can be translated into clinical practice? Specifically, in patients who might not be able to perform cognitive tasks, how would this impact the use of EEG for measuring consciousness?
    1. As mentioned in the introduction (lines: 60-64), how cognitive load affects proposed markers of consciousness is important first and foremost when considering potential sources for error when inferring the status of the patient. For example, if a patient is deaf (unbeknownst to the doctors) then an auditory paradigm is unsuited. For cognitive load specifically, a non-responsive patient in delirium may not attend to a cognitive task such as the P3b (essentially mind-wander), or a patient that doesn’t speak the language may perform some other unrelated cognitive task not specified by the doctors (e.g. trying to figure out a mathematical problem). In short, if a diagnostic tool depends on a patient following instructions (be it to relax or to perform some task) then it may be confounded by various factors not under the doctors’ control. Clinically, this is problematic, especially in patient populations whose cognitive and/or sensory capacity is temporarily or permanently reduced. Lines 552-553 and 540-542 highlight this fact; a measure of consciousness used in the clinic should not be dependent on task compliance. To answer the question, this study suggests that the metrics PCI, LZc, ACE, SCE, ought to be considered over measures that are associated with a given task such as the P3b. We feel that the conclusion, as written, supports this. 

Minor comments:

  1. In the sentence "Several methods aiming at objectively quantifying the presence or degree of consciousness have been developed", the phrase "aiming at" should be rephrased to "aiming to" for better clarity.
    1. Thank you. This has now been fixed.

Round 2

Reviewer 1 Report

Comments and Suggestions for Authors

The author has provided a satisfactory response to the raised issues, and it is recommended for publication.

Reviewer 2 Report

Comments and Suggestions for Authors

I thank the authors for their rebuttal. I don't have any further comments.